# Efficacy of imaginative vocal training for enhancing vocal awareness in non-professional singers

Federica Biassoni[ID][1,2,3]*, Giulia Vismara[4], Martina Gnerre[ID][4]

1 Research Center in Communication Psychology, Catholic University of the Sacred Heart, Milan, Italy,
2 Music Psychology Research Unit, Catholic University of the Sacred Heart, Milan, Italy, 3 Department of Psychology, Università Cattolica del Sacro Cuore, Milano, Italy, 4 Communication Psychology Lab, Faculty of Psychology, Catholic University of the Sacred Heart, Milan, Italy

* federica.biassoni@unicatt.it

## Abstract

The objective of this study was to examine whether different types of mental-imagery training focused on the vocal apparatus can enhance awareness of the vocal tract and diaphragm (vocal awareness) in non-professional singers.Sixty participants with no singing education received one of three training conditions: following instructions based on 1) a description of the physiological changes that take place during phonation (physiological description), 2) imitating an action using the vocal apparatus (imitative action), and 3) a metaphorical narration. Imitative action and metaphorical narration were conceptualized as more imaginative forms of training. Vocal awareness was assessed with a questionnaire that participants completed before and after the training. The questionnaire measured three indices: vocal apparatus representation, vocal apparatus interoceptive awareness, and vocal self-regulation. Results showed that all three types of training program significantly enhanced vocal awareness, but imitative action and metaphorical narration were more effective for interoceptive awareness, and metaphorical narration was more effective for self-regulation. In conclusion, the two imaginative forms of training were more effective than physiological description for improving vocal awareness.

## Introduction

Knowledge of the body, and of the voice, is of the utmost importance when learning to sing, as singers are their own instrument [1]. Vocal awareness, an essential element of the vocal self-concept, encompasses knowledge and understanding of one's own unique voice [1–3]. It demands a mindful grasp of the processes facilitated by the muscles of phonation that generate and modify the voice [4]. This understanding serves as the foundation for the most proficient use and control of the voice. The limited and fragmented body of literature on enhancing vocal awareness converges

**Data availability statement:** Data Availability Statement: The data that support the findings of this study are available at https://osf.io/aqfz4/?view_only=1a64cdaaab3e-4fa69a5f4b273d33d7a3.

**Funding:** The author(s) received no specific funding for this work.

**Competing interests:** The authors have declared that no competing interests exist.

on three key abilities: (1) vocal apparatus representation, (2) developing interoceptive awareness of the vocal apparatus, and (3) exercising vocal self-regulation [5–14]. Proponents of the Feldenkrais method [10], the Linklater method [12], and the Magnani method [11] argue that attending to one's own voice and learning through the direct experience of bodily sensations foster greater awareness of its power and qualities. Similarly, within the Borragán method, Agudo et al. [7,8] emphasize the importance of body intelligence, that is, the ability to represent and remain aware of one's own posture and actions. In contrast, teachers of the Wilfart [15], Le Huche [5,6], and Weiss [14] methods maintain that there is a relationship between the perception and representation of subjective bodily experience, on the one hand, and the effective management of vocal performance, on the other.

Although these methods differ in their specific procedures and theoretical underpinnings, they converge in emphasizing the central role of bodily awareness, and particularly vocal awareness, in supporting performance in both speech and singing. Within the vocal pedagogy literature [16–21], the vocal tract and the diaphragm are consistently identified as key components of the vocal apparatus, and of the phonatory system more broadly, that warrant particular attention when cultivating bodily awareness. It has been proposed that variability in vocal performance depends in part on the degree of awareness of these components, even though they are often poorly understood by learners and performers [22,23]. It has also been proposed that mental imagery contributes to vocal awareness by enabling individuals to form and draw on mental representations of the anatomical structures involved in vocal production and how they move [24,25]. For example, mentally rehearsing the movements of the articulators involved in speech production, and anticipating the resulting auditory feedback, can be effective because it engages both the motor and perceptual systems [26].

Vocal awareness is as important for non-professional singers as it is for those who earn their living by using their voices in a professional capacity. Moreover, increasing awareness of the body, and particularly of the voice, generally enhances self-knowledge [27–29] and can improve the effectiveness with which emotions and other communicative intentions are conveyed [30–32]. Although several methods for teaching singing are grounded in bodily knowledge and vocal awareness [7–10,12,13,33–35], relatively few studies have evaluated their effectiveness [7,8,27,36], and even fewer have focused on their application with singers who do not aspire to professional careers. Evidence supports the effectiveness of mental-imagery training to improve the performance of athletes [37] and actors [38], but not in singers. Knowledge and awareness of the body – especially the vocal apparatus – are as important to singers as they are to athletes and actors. We were interested in the potential for using mental imagery in vocal training. We therefore investigated the effects of three types of training based on mental imagery on non-professional singers' vocal awareness. We chose to focus on awareness of the vocal tract and diaphragm because these are the most important components of the vocal apparatus. Singers rely on mental representations of their body and specifically their vocal apparatus to control phonation, so it is essential that they can form and draw on these mental representations [39–42].

## Factors involved in vocal awareness

Many studies have shown that people who attend to their own voluntary and involuntary bodily actions and responses can enhance their knowledge of the body and increase their motor-control efficiency [43–49]. We have already referred to three key abilities underlying vocal awareness: 1) the vocal apparatus representation; 2) interoceptive awareness of the vocal apparatus; and 3) self-regulation, which involves managing and controlling vocal behaviors—such as pitch, volume, quality, and resonance—through conscious monitoring, adjustment, and refinement of vocal techniques and habits.

To acquire adequate motor-perceptual knowledge of the functioning of the vocal system, one must be able to perceive the actions of its different parts [44,50–53] and to picture them mentally [46–49,51,54–56]. To be able to represent one's own voice (i.e., to be aware of it and listen to it internally) one must process and make use of physical sensations in the body [44,57–60], especially the functions of its internal organs, such as blood pressure, heartbeat, thermoregulation, breathing rhythm, digestion, and feelings such as arousal, which is related to the activity of the autonomic nervous system [57,58,61,62]. Together, these are known as interoception. The perception of muscle tension, movement, posture, balance, and tendon joint activity [44,45,63] is known as proprioception [7–10,14]. It has been suggested that the mouth and the larynx contain proprioceptive receptors that aid their localization and muscle control [16–20], but research on the proprioceptive experience of the diaphragm has yielded inconclusive and disputed findings [17,21].

It has been hypothesized that, to self-regulate and modulate vocal performance, one must be able to mentally represent the vocal apparatus, including the phonatory system, to use interoceptive and proprioceptive awareness of the vocal apparatus in action, and to integrate this awareness and knowledge with executive control and emotional sensitivity [7,8,10,64,65]. Methods of teaching singing based on these hypotheses have been tested and shown to be effective [5–8,11,13,14].

## Mental imagery and vocal awareness

One way of teaching singing is using the so-called empirical method [34,35,66]. This method uses language to evoke images that enable the singer to prepare themself both physically and mentally to sing [5,6,11–13,67]. Phonation can be activated in two ways: via imitative action or responding to a metaphorical narration, both of which require imagination on the part of the singer. Imitative action relies on analogical thinking, whereby parallels or similarities are drawn between two or more concepts, situations, or objects to understand or solve problems, while metaphor is a form of symbolic thinking, referring to symbols that may not be physically present [79]. Imitative action facilitates learning and understanding by encouraging learners to replicate observed behaviors and to create a connection between a model and the action to be performed, whereas metaphorical narration creates connections between concepts. Imitative action and metaphorical narration allow learners to gain a nuanced understanding by making connections between singing and familiar concepts. In short, they can both serve as cognitive strategies used to grasp and reproduce phonatory acts.

In imitative action training, learners are asked to create a mental image of how the vocal apparatus moves when they speak by simulating actions (doing); in metaphorical narration training they imagine carrying out particular behaviours that can affect the way they speak (imagining) [5,6,11–13,34,35,66,67]. Both imitative-action and metaphorical-narration training are forms of mental-imagery training that require imagination and draw on the connection between the use and understanding of abstract and metaphorical language and sensorimotor bodily experience [68–72]. Several studies have shown that metaphorical language may help one think about and explore one's own body and its functions more easily [73–81].

## Objectives

In this study we tested the effects of three training programs on three indices of vocal awareness collected through a self-report questionnaire completed before (Time 1) and after the training (Time 2). The three training programs required participants to follow instructions based on 1) a description of the physiological changes that take place during phonation

(physiological description), 2) imitating an action using the vocal apparatus (imitative action), and 3) a metaphorical narration. The three indices of awareness of the vocal tract and diaphragm respectively were 1) vocal apparatus representation2) interoceptive awareness of the vocal apparatus (awareness), and 3) vocal self-regulation (self-regulation).

## Hypotheses

Based on the reviewed literature, we developed three key hypotheses to guide our investigation. First, we anticipated a general effect of time on vocal awareness (H1). Specifically, we expected that participants, regardless of the training method they received, would exhibit higher vocal awareness scores after training compared to before. Second, we hypothesized that the type of training would play a significant role in shaping vocal awareness (H2). More specifically, we predicted that participants undergoing imaginative training—whether through imitative action or metaphorical narration—would demonstrate greater improvements in vocal awareness compared to those receiving physiological description training. Finally, we expected an interaction effect between time and training type (H3). In other words, we hypothesized that the increase in vocal awareness from pre- to post-training would be more pronounced among participants exposed to imaginative training (imitative action or metaphorical narration) compared to those who received training based on physiological descriptions.

## Method

### Participants

Sixty non-professional singers (30 F, 30 M) took part in the study. Their mean age was 30.5 years (SD = 7.54, range 19–45). The size of the sample was calculated using G*Power 3.1.9.2 [82]. To achieve a statistical power of 95%, with a medium effect size (0.25), and an α level of.01, the sample size of 60 is recommended. Participants were recruited through advertisements in the local media. Inclusion criteria were that: 1) participants had not attended any vocal course or training focused on vocal awareness nor singing courses before, and 2) participants had no vocal nor hearing impairment. Participants were divided into three groups of 20 matched for sex and age and then assigned quasi-randomly to a group in which they received one form of training (physiological description, imitative action, or metaphorical narration) to enhance awareness of the vocal tract and diaphragm.

Participants were recruited between November 15, 2023, and February 10, 2024. All participants provided written informed consent prior to taking part in the study. The research was approved by the Ethics Committee for Research in Psychology (CERPS) of the Catholic University of the Sacred Heart (Protocol No. 60−23, Approval Date: May 24, 2023). No minors were involved in the study.

### Materials

**My voice and me questionnaire.**  Since no questionnaire measuring vocal awareness has been published, we developed an ad hoc questionnaire. First, the questionnaire measured participants' ability to form avocal apparatus representation. Items 1, 4, and 8 were based on previous research [39–42], adapted from the Body Consciousness Questionnaire [83] and the Body Intelligence Scale [84] with an α of.53. In Item 8, participants were shown the silhouette of a human being and asked to locate the vocal apparatus on the silhouette; correct and incorrect responses were identified by means of a numbered grid. Second, it measured participants' interoceptive and proprioceptive awareness of the vocal apparatus. Items 2, 5, and 7 were based on previous research [7–10,14] with an α of.75. Third, it measured participants' vocal self-regulation. Items 3 and 6 were adapted from a voice perception questionnaire reported in Fussi [85] and the Voice Handicap Inventory [86] with an α of.74. Three experts in singing and voice education rated the comprehensibility, relevance, and consistency of the items testing the three forms of vocal awareness, and 10 participants in a pilot study (5 m, 5 f), aged 20–40 years, also rated the comprehensibility of the

questions and responses. Other than for the item for which participants located the vocal apparatus, responses were given using 5-point Likert-type scales from 1 (strongly disagree) to 5 (strongly agree), so the maximum that could have been scored for the seven scored items was 35.

## Mental-imagery training programs

Each of the three training programs was designed to be consistent with the literature on singing teaching [5–8,11–13,34,35,87,88] and focused on awareness of the vocal tract and diaphragm.

**Physiological description.** According to Garcia [89] specialized singing training requires an in-depth understanding of vocal physiology, learned from description and practice [11,90]. Physiological descriptions are technical explanations of the functions of the vocal tract and diaphragm and may include instructions for specific exercises, such as "Move the base of your tongue back;" "Lower your diaphragm;" "Contract your pelvic floor" [11]. This was the most concrete of the three training programs in its use of descriptive, specialist terminology and required the participant to make least use of their imagination.

**Imitative action.** According to Mercado et al., [91], vocal imitation involves synchronizing an auditory stimulus with the vocal motor system, whether this is achieved with or without effort. Rather than asking participants to imitate an auditory stimulus we asked them to act as if they were carrying out a task that would influence phonation, such as "Sing as if you were yawning;" "Sing as if you had a hot potato in your mouth" [34,35]. If the participant followed the instruction correctly, they would produce, more or less automatically, the correct position of tongue, mouth, and palate for singing (vocal posture). This training program was more abstract than physiological description in its use of language and required the participant to make more use of their imagination.

**Metaphorical narration.** Participants were invited to listen to narratives evoking images to illustrate the physiological processes that take place in the vocal system, and to use mental imagery themselves while experiencing the physical sensations of speaking. The processes of breathing in and out, for example, are evoked in the instruction "Speak as if an umbrella was opening inside your belly," which could raise the participant's awareness of their abdominal muscles and thus help them lower their diaphragm, Similar instructions include "Speak as if the sound were coming out of your back;" "Smile internally when you sing;";"Feel as though your body were a sandbag leaning against a marble column" [12,34,35]. Damasio [41] also emphasizes the importance of iconic and symbolic language in the representation of body, such as the limbs and parts of the vocal apparatus, including the phonatory system, and actions; he describes our capacity for creating mind maps, whether or not these are in the form of images, that help us imagine and represent the shape of our limbs and their position in space. This was the most abstract of the three training programs in its use of metaphorical language and required participants to make the most use of their imagination. See Table 1 for a summary of the three programs.

## Procedure

Participants undertook single experimental sessions, individually, in a soundproofed room. They were asked to stand up while they were carrying out the pre- and post-training performance tasks, but they sat down while completing the pre- and post-task questionnaires and while receiving the training. They carried out each of the performance tasks three times.

The protocol was as follows for vocal tract awareness:

1. Pre-training performance task (5 s): utter the phrase Ciao Aurora(using resonators and vowel articulation to produce appropriateprosody)

2. Complete My Voice and Me questionnaire

3. Training phase (5 min)

**Table 1. Summary of training programs and the instructions for developing vocal awareness.**

| Training program | Description | Vocal tract awareness | Diaphragm awareness |
|---|---|---|---|
| Physiological description | Aims to teach an understanding of vocal physiology by describing it in specialist terms and giving instructions for using the vocal apparatus. | Move the base of your tongue back to lower your larynx, then lift your soft palate to lengthen the oropharyngeal space. This will produce a larger oral space and a wider pharyngeal and vestibular space. Now repeat the phrase *Ciao Aurora* keeping your vocal apparatus in the same position. | Breathe in deeply by contracting your diaphragm so that your abdominal wall should be engaged. In order for your diaphragm to stabilize, make sure it exerts a force directly proportional to the one exerted by your pelvic floor in the opposite direction. You should also lean on the support of the abdominal muscles by slightly contracting your pelvic floor. Utter the sound *ee* and hold it until you run out of breath. |
| Imitative action | Aims to encourage participants to produce sounds by imitating an action that influences phonation, even if it is not directly connected to it. | Imagine singing as if you had an egg in your mouth slowly descending through your throat. Repeat the phrase *Ciao Aurora* while keeping the egg inside your mouth; avoid crushing or breaking it. | Think of wearing a lifebelt around your waist, one size larger than your actual size. Visualize it and try to keep it balanced by extending your hips and belly. Hold this position and say *ee* until you run out of breath. |
| Metaphorical narration | Aims to illustrate the changes that occur in the body during voice production using a symbolic narrative. | Imagine your mouth as a meadow in the mountains and your tongue as a delightful small lake. The sun is warming the lake. This scenery is bringing you great harmony.<br>At a certain point, some soap bubbles start arising from the lake. One of these bubbles is settling on your tongue attachment and begins to grow slowly. The hills and mountains situated along the perimeter of your mouth are extending and moving to make room for the slowly and steadily growing bubble. They are moving aside gently to avoid any sudden movements that might pierce the sides of the bubble. The bubble is growing and rising more and more until it can touch your palate and tongue. Repeat the phrase *Ciao Aurora* keeping this image in your mind. | Imagine that your rib cage is expanding more and more because elastic bands attached to it are being pulled steadily outwards and downwards.<br>Visualize the interior of your rib cage interior as if it were a large empty pit that is gradually expanding. Focus on the feeling of pulling the bands outwards and downwards and by the pressure caused by the expansion of the pit, then say *ee* until you run out of breath. |

4. Post-training performance task for vocal tract awareness (5 s): utter Ciao Aurora again

5. Complete My Voice and Me questionnaire again.

The protocol was as follows for diaphragm awareness:

1. Pre-training performance task: maintain the vowel sound ee for as long as possible (a vocalization depending on the proper management of breath and breathing)

2. Complete My Voice and Me questionnaire

3. Training phase (5 min)

4. Post-training performance task for diaphragm awareness: maintain thevowel sound ee for as long as possible again

5. Complete My Voice and Me questionnaire again.

## Data analyses

Mixed-model 2 x 3 analyses of variance (ANOVA) were conducted. The within-participant variable was time (pre- vs. post-training) and the between-participants variables were the three training programs (physiological description, imitative

action, metaphorical narration) for each of the three outcome measures (vocal apparatus representation, interoceptive awareness, and vocal self-regulation) relating to the vocal tract and diaphragm tasks respectively. Tukey post hoc analyses were conducted. The level of statistical significance was set at p ≤ .05. Bonferroni corrections were applied.

## Results

Mean and standard deviation values for the training activities are reported in Table 2. We predicted a main effect of time, such that participants would score higher on measures of vocal awareness after taking the training, and a main effect of training type, such that participants would score higher on measures of vocal awareness (both for the diaphragm and vocal tract) following imaginative training, either through imitative action or metaphorical narration.

### Vocal apparatus representation

For vocal tract, the analysis revealed a significant main effect of time, [$F(1, 57) = 10.21$, $p < .001$, $\eta^2_p = .15$], indicating a decrease in vocal apparatus representation scores from pre-training ($M = 6.97$, $SD = 1.76$) to post-training ($M = 6.73$, $SD = 1.91$). This is inconsistent with H1. Moreover, no significant main effect of training type was found, suggesting that the three training methods did not differ significantly in their impact on vocal apparatus representation scores ($p > 0.05$). Moreover, no significant interaction effect was found ($p > .05$). Thus, H2 and H3 were not supported.

A chi-square test assessed changes in participants' identification of phonatory parts before and after training, based on their selections on a human figure (Table 3). The results showed a statistically significant difference for several phonatory parts: Heart: $\chi^2(1) = 5.84$, $p = .016$; lungs: $\chi^2(1) = 9.17$, $p = .002$; head: $\chi^2(1) = 33.22$, $p < .001$; mouth (tongue, palate, lips): $\chi^2(1) = 7.80$, $p = .005$; diaphragm: $\chi^2(1) = 23.47$, $p < .001$. Conversely, no statistically significant difference was found for the vocal cords and throat, although a trend was observed ($\chi^2(1) = 3.15$, $p = .076$). Specifically, after the training, the mouth

**Table 2. Mean and standard deviation values are reported for the training activities.**

| Condition | Pre-training (diaphragm) | Post-training (diaphragm) | Pre-training (vocal tract) | Post-training (vocal tract) |
|---|---|---|---|---|
| | Mean (SD) | Mean (SD) | Mean (SD) | Mean (SD) |
| Training 1 (PD) | | | | |
| VAR | 6.45 (1.96) | 7.05 (1.63) | 6.45 (2.08) | 6.2 (2.14) |
| VAIA | 8.8 (2.82) | 11 (2.90) | 8.8 (2.82) | 9.55 (3.24) |
| VSR | 6.95 (2.26) | 6.3 (2.56) | 6.95 (2.26) | 6.25 (2.66) |
| All PD | 22.20 (1.80) | 24.35 (1.85) | 22.20 (1.80) | 22.00 (1.85) |
| Training 2 (IA) | | | | |
| VAR | 7.25 (1.58) | 7.85 (1.49) | 7.25 (1.57) | 7.1 (1.65) |
| VAIA | 10.3 (2.79) | 12.3 (2.26) | 10.3 (2.79) | 11.5 (2.48) |
| VSR | 7.25 (1.39) | 6.9 (1.84) | 7.25 (1.39) | 6.45 (1.64) |
| All IA | 24.80 (1.85) | 27.05 (1.90) | 27.05 (1.90) | 27.05 (1.90) |
| Training 3 (MN) | | | | |
| VAR | 7.2 (1.64) | 8.15 (1.41) | 7.2 (1.64) | 6.9 (1.94) |
| VAIA | 10.90 (1.83) | 12.7 (1.92) | 10.9 (1.83) | 11 (2.73) |
| VSR | 7.8 (1.57) | 8.05 (1.83) | 7.8 (1.32) | 6.8 (1.99) |
| All MN | 26.00 (1.95) | 26.00 (1.95) | 26.00 (1.95) | 26.00 (1.95) |
| All imaginative | 50.80 (3.12) | 55.95 (3.22) | 50.70 (3.00) | 50.70 (3.00) |
| Mean imaginative | 25.40 (1.44) | 27.98 (1.58) | 25.35 (1.82) | 24.83 (1.48) |

*Note.* PD = Physiological description, IA = imitative action, MN = metaphorical narration, VAR = vocal apparatus representation, VAIA = vocal apparatus interoceptive awareness (VAIA), VSR = and vocal self-regulation.

**Table 3. Chi-Square Test Results for the Identification of Phonatory Parts Before and After Training.**

| Body Part | $\chi^2$ | p |
|---|---|---|
| **Vocal Apparatus Representation** | | |
| **Vocal Tract Awareness** | | |
| Heart | 5.84 | .016 |
| Lungs | 9.17 | .002 |
| Head | 33.22 | <.001 |
| Mouth (tongue, palate, lips) | 7.80 | .005 |
| Diaphragm | 23.47 | <.001 |
| Vocal Cords & Throat | 3.15 | .076 |
| **Diaphragm Awareness** | | |
| Vocal Cords & Throat | 77.91 | <.001 |
| Heart | 80.53 | <.001 |
| Lungs | 82.65 | <.001 |
| Head | 101.40 | <.001 |
| Diaphragm | 74.53 | <.001 |

was more frequently recognized as a key part of the vocal tract responsible for producing vocal utterances than before the training (see Fig 1), and subjects who underwent imitative action training identified the throat and mouth as the origin of voice production less frequently in their post-training performance than in their pre-training performance (throat: 90% in pre-training performance vs. 70% in post-training performance; mouth: 30% in pre-training performance vs. 15% in post-training performance). In contrast, subjects who participated in physiological description training identified the throat and mouth as the origin of voice production more frequently in their post-training performance than in their pre-training performance (throat: 65% in pre-training performance vs. 70% in post-training performance; mouth: 20% in pre-training performance vs. 30% in post-training performance).

As far as diaphragm representation was concerned, no significant main effect of time emerged (p > .05); therefore, H1 was not supported. However, there was a significant main effect of training type, $F(2, 57) = 24.36$, $p < .001$, $\eta^2_p = .30$. Post hoc analyses showed that participants in the imitative action training (M = 7.85, SD = 1.49) and metaphorical narration

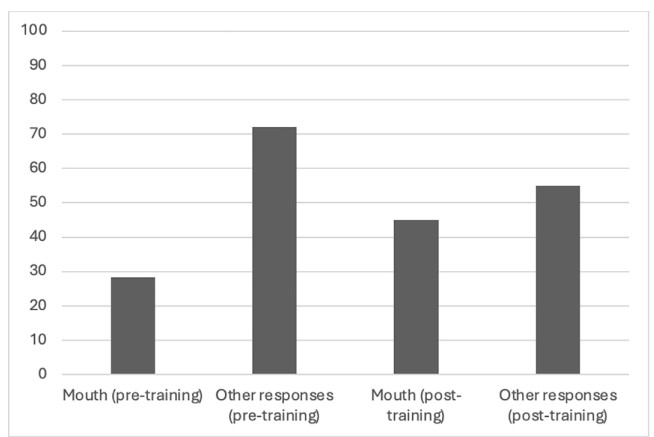

**Fig 1. Percentage of responses provided when requested to specify the body parts where voice originates (vocal tract awareness): Mouth (pre-training): 28%; Other responses (pre-training): 72%; Mouth (post-training): 45%; Other responses (post-training): 55%.].**

training (M = 8.15, SD = 1.41) conditions reported significantly higher diaphragm representation scores than those in the physiological description condition (M = 7.05, SD = 1.63), p < .05, supporting H2. No significant time × training interaction emerged (p > .05), indicating that the effect of training did not vary over time; thus, H3 was not supported. A chi-square test was conducted to compare participants' identification of different phonatory parts before and after training, based on their selections on a human figure (Table 3). The results indicated a statistically significant difference for several phonatory parts: Vocal cords and throat: $\chi^2(1)$ = 77.91, p < .001; Heart: $\chi^2(1)$ = 80.53, p < .001; Lungs: $\chi^2(1)$ = 82.65, p < .001; Head: $\chi^2(1)$ = 101.40, p < .001; Diaphragm: $\chi^2(1)$ = 74.53, p < .001. However, the diaphragm was identified as the main part involved in the proposed vocal exercise in the post-training phase (55%), unlike the pre-training phase, where this muscle was given little relevance (25%) (see Fig 2).

### Vocal apparatus interoceptive awareness

Regarding the vocal tract, participants demonstrated significantly higher interoceptive awareness in the post-training performance compared to the pre-training performance [$F(1, 57)$ = 5.77, $p$ = .05, $\eta^2_p$ = .021], confirming H1. Specifically, participants reported significantly higher scores in the post-training phase (M = 10.68, SD = 2.82) compared to the pre-training phase (M = 10.00, SD = 2.48), p = .04. A significant main effect of training was also found [$F(2, 57)$ = 3.96, p = .04, $\eta^2_p$ = .056], indicating that training type influenced interoceptive awareness scores, confirming H2. Post hoc analyses revealed that participants in the imitative action training group demonstrated significantly greater improvement in interoceptive awareness compared to those in the physiological description training group (p < .001). However, no significant difference was found between the Imitative Action and metaphorical narration training groups (p > .05). A significant time × training interaction effect was also found [$F(2, 57)$ = 3.42, $p$ = .05, $\eta^2_p$ = .03], confirming H3. This suggests that the effect of training on interoceptive awareness varied over time. Post hoc analysis for the interaction effect confirmed that interoceptive awareness scores significantly increased from pre-training (M = 10.3, SD = 2.79) to post-training (M = 12.3, SD = 2.26) in the imitative action training group, p < .05, and from pre-training (M = 10.9, SD = 1.83) to post-training (M = 12.7, SD = 1.92) in the metaphorical narration training group, p < .05. However, no significant improvement was observed in the physiological description training group, p > .05. These findings confirm that imaginative training methods (IA and MN) were more effective than PD in enhancing interoceptive awareness, both in overall comparison and in pre-post changes, supporting H1 and H2.

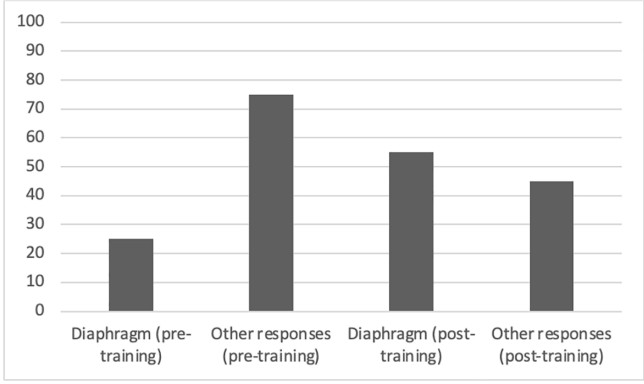

**Fig 2. Percentage of responses provided when requested to specify the body parts where voice originates (diaphragm awareness): diaphragm (pre-training): 25%; Other responses (pre-training): 75%; diaphragm (post-training): 55%; Other responses (post-training): 45%.].**

Regarding the diaphragm, participants reported significantly higher interoceptive awareness scores in the post-training phase (M = 12.00, SD = 2.36) compared to the pre-training phase (M = 10.00, SD = 2.48), F(1, 57) = 39.95, p < .001, $\eta^2_p$ = .41, confirming H1. The main effect of training type was also significant, F(2, 57) = 11.06, p < .001, $\eta^2_p$ = .28, indicating that training condition had a substantial impact on diaphragm interoceptive awareness, thus supporting H2. Post hoc analyses showed that participants in the imitative action training group (M = 12.30, SD = 2.26) and in the metaphorical narration training group (M = 12.70, SD = 1.92) reported significantly higher scores than those in the physiological description group (M = 11.00, SD = 2.90), p < .05, whereas no significant difference emerged between the imitative action and metaphorical narration groups (p > .05). The time × training interaction was not significant, F(2, 57) = 0.33, p = .72, $\eta^2_p$ = .01, indicating that the pattern of pre- to post-training change did not differ reliably across training conditions; therefore, H3 was not supported, and any apparent differential gains should be interpreted with caution.

### Vocal self-regulation

Regarding the vocal tract, participants reported significantly lower vocal self-regulation scores in the post-training phase (M = 6.80, SD = 2.09) compared to the pre-training phase (M = 7.33, SD = 1.65), F(1, 57) = 5.68, p = .021, $\eta^2_p$ = .09, contradicting H1. The main effect of training type was not significant, F(2, 57) = 1.31, p = .28, $\eta^2_p$ = .02, indicating that the three training methods did not differentially affect overall vocal self-regulation scores; thus, H2 was not supported. Although pairwise comparisons suggested slightly higher self-regulation scores in the imitative action (M = 7.25, SD = 1.39) and metaphorical narration (M = 6.80, SD = 1.99) groups compared to the physiological description group (M = 6.25, SD = 2.66), p = .043, these differences should be interpreted with caution given the non-significant omnibus effect. The time × training interaction was not significant (p > .05), indicating that the pattern of pre- to post-training change did not differ reliably across training conditions; therefore, H3 was not supported for vocal self-regulation of the vocal tract. Regarding the diaphragm, participants reported significantly lower vocal self-regulation scores in the post-training phase (M = 7.08, SD = 2.07) compared to the pre-training phase (M = 7.33, SD = 1.74), F(1, 57) = 5.19, p = .027, $\eta^2_p$ = .08, again contradicting H1. In contrast, the main effect of training type was significant, F(2, 57) = 11.06, p < .001, $\eta^2_p$ = .16, indicating that training condition substantially influenced diaphragm-related self-regulation, supporting H2. Post hoc analyses showed that participants in the metaphorical narration group (M = 8.05, SD = 1.83) reported higher diaphragm self-regulation scores than those in the physiological description and imitative action groups (ps < .05). The time × training interaction was not significant, F(2, 57) = 0.33, p = .72, $\eta^2_p$ = .01, indicating that the effect of training on diaphragm self-regulation did not vary significantly over time; therefore, H3 was not supported.

### Discussion

This study investigated the impact of three training approaches—physiological description, imitative action, and metaphorical narration—on vocal awareness, specifically focusing on vocal tract and diaphragm perception. Vocal awareness was assessed using three indices: vocal apparatus representation, vocal apparatus interoceptive awareness, and vocal self-regulation. Although not all measures showed a significant and consistent effect of training across indices of vocal awareness, overall the findings indicate that imaginative training methods (imitative action and metaphorical narration) significantly influenced the development of vocal awareness.

Importantly, training based on imitative action and metaphorical narration was more effective than physiological description in enhancing vocal awareness on most indices. This pattern of findings supports H2 and is consistent with previous research indicating that symbolic imagery and metaphorical language foster deeper and more integrated mental representations of the body [72,76,81]. Furthermore, vocal apparatus interoceptive awareness emerged as the most responsive index across both the vocal tract and diaphragm conditions, reinforcing the idea that interoceptive awareness is a key factor in vocal training and self-perception. Although counter-intuitive, the post-training decreasing in measures of vocal apparatus representation and vocal self-regulation suggests that an early "conscious incompetence" phase of

skill acquisition may arise from the training. Indeed, the intervention may have enhanced metacognitive resolution, as indicated by gains in interoceptive awareness, prompting participants to apply stricter internal criteria to what counts as an accurate representation of their vocal apparatus and as effective self-regulation. In this view, lower self-ratings may reflect a response-shift (criterion tightening), rather than a decrease in the participants' skills: it is plausible that they moved from a coarse, overconfident baseline to a more realistic appraisal. This interpretation of their skills is in line with the Response Shift Theory [92]. Specifically, our findings suggest that the imaginative training appears to have acted as a catalyst, triggering a recalibration of internal standards [92, p. 32], whereby participants began to judge their abilities against a more refined and demanding scale. This recalibration may be looked at as a positive, adaptive process that signifies a deeper engagement with the nuances of vocal production. Consequently, the initial decline in self-reported scores can be interpreted as a necessary and valuable step in the learning curve. It marks the transition from a vague, undifferentiated understanding of one's voice to a more discerning self-awareness, which forms the essential foundation upon which more advanced vocal technique and reliable self-regulation can be built in future training.

## Vocal awareness of vocal tract

In terms of vocal apparatus representation, no significant effect of time or training was observed, suggesting that none of the training methods significantly altered participants' explicit representation of the vocal apparatus and that neither hypothesis was supported. The representation of the vocal apparatus may involve a conscious cognitive process that is relatively stable over time and may not be easily modified through short-term training [93].

For vocal apparatus interoceptive awareness, a significant main effect of time and training was found, corroborating both hypotheses. This suggests that imaginative techniques facilitate a deeper understanding of the internal bodily processes related to phonation. These findings align with the existing literature [13,25,46,47,94], which emphasizes that bodily imagery and mental representation can enhance bodily perception and interoceptive awareness. In terms of vocal self-regulation, the results did not support an overall improvement in self-regulation, as post-training scores were lower than pre-training scores, contrary to H1. We interpreted this result considering that vocal regulation is a metacognitive process beyond basic bodily awareness, requiring the ability to adapt and modulate voice use based on situational demands. In other words, we suggest that improvements in vocal regulation do not always translate into an immediate sense of control. Instead, they may initially increase awareness of errors and difficulties in voice modulation. However, training type did influence vocal self-regulation, consistently with H2. Participants in the imitative action and metaphorical narration groups indeed exhibited higher scores compared to those in the physiological description group. This suggests that methods relying on imagination and embodied experience may be more effective in fostering self-regulation abilities than those focused solely on explicit anatomical descriptions. This is supported by the theory of embodiment [95], which may suggest that active involvement of the body and imagination enhances the ability to regulate one's vocal actions more fluidly and intuitively.

## Vocal awareness of diaphragm

Regarding diaphragm awareness, training type significantly influenced vocal apparatus representation, with imitative action and metaphorical narration leading to greater improvements than physiological description. These results support H2, reinforcing the idea that more imaginative and experiential learning methods are more effective in fostering awareness of the vocal apparatus. Unlike for vocal tract awareness, however, not all the measures consistently indicated a significant effect of time, suggesting that improvements in diaphragm representation were primarily driven by training type rather than general exposure or repetition. For interoceptive awareness of the diaphragm, the results confirmed both H1 and H2, with significant improvements following training. The strongest effects were observed for metaphorical narration and imitative action, further highlighting the efficacy of these training methods in enhancing interoceptive abilities. These training methods may have facilitated a more vivid and accessible mental representation of diaphragm dynamics, making

the perception of its activation more intuitive. Such result is consistent with previous studies on body representation, which highlight how the use of mental imagery and symbolic narratives can enhance awareness of body parts that may also not be immediately visible [96–98].

In terms of vocal self-regulation related to the diaphragm, on the other hand, the findings did not support H1, as post-training scores were lower than pre-training scores. However, H2 was confirmed, since the metaphorical narration group showed higher scores in the post-training phase compared to the pre-training phase. This suggests that the use of metaphorical imagery may help participants internalize self-regulation strategies more effectively than physiological description or imitative action. Our interpretation is that the symbolic language used in metaphorical narration helps foster greater conscious control of the voice, particularly in managing breathing and diaphragmatic support [5,6,11–13,34,35,66]. These results may be explained by the ability of metaphors to bridge abstract concepts with body sensations and diaphragm-related processes, as observed in studies investigating the role of metaphorical language in improving body awareness [73,98,99].

## Conclusions

These findings emphasize the potential value of implementing imaginative techniques in vocal education programs to foster a deeper understanding of the physiological aspects of voice production and of the embodied inner perception related to vocal awareness. Notably, the study suggests that training programs that use non-technical language could be especially beneficial for non-professional singers. This opens the possibility of deploying this type of vocal-awareness training to strengthen awareness and the ability to mindfully use one's vocal apparatus in different groups of people, not only those interested in singing but also professionals whose work relies on vocal performance, including actors, call-center operators, teachers, and radio operators. In other words, on the one hand this approach can democratize the access to effective vocal training, making it more approachable and understandable for individuals without a technical background in music or voice. At the same time, imaginative training designed to enhance vocal awareness may offer an opportunity not only to develop more effective vocal skills, but also to cultivate broader abilities related to bodily awareness, self-contact, and the capacity to express oneself and communicate effectively. Though the conclusions from this research provide some exciting insights to develop training for vocal education based upon the empowerment of awareness of the singing process, some limits merit being noted. Firstly, the My Voice and Me self-report questionnaire has not yet been validated. However, it was developed based on established theoretical frameworks and expert input, ensuring face validity. Additionally, future research should incorporate a more diverse sample to represent the full spectrum of expertise levels. For instance, examining the effectiveness of these training programs in both professional and non-professional singers would require a larger sample and constitutes a promising direction for future studies.

A key limitation concerns the low internal consistency of the Representation subscale. We observed a relatively low Cronbach's alpha ($\alpha = .53$). However, this is not unexpected, given that the subscale comprises only three items. Alpha is sensitive to both scale length and average inter-item covariance and tends to increase with the number of items, and for very short scales the coefficient may underestimate internal consistency [100–102].

Furthermore, the questionnaire assumes a direct correlation between body and voice; however, this connection may not naturally emerge in non-professional singers. Future research could further investigate vocal awareness by integrating physiological measurements to corroborate self-reported data, providing a more comprehensive understanding of the interplay between body and voice.

Additionally, while the study indicated improvements in vocal awareness and technique within the duration of the training, the durability of these effects beyond the study's timeframe remains uncertain. Long-term follow-ups to assess the retention and practical application of the acquired vocal skills would provide a more comprehensive understanding of the training's lasting impact. Lastly, the discrepancy in the length (number of words) of instructions among the three

conditions, as well as the varying cognitive complexity of the mental operations required, may have influenced the study results and deserves further discussion.

As far as future directions for this research are concerned, it would be meaningful to compare the perception of increased awareness self-reported by the participants and the actual differences of the acoustic features of the vocal performance, which were not considered in this study, as well as to develop further instruments to measure the subjective perception of the change in vocal awareness.

## Author contributions

**Conceptualization:** Federica Biassoni, Giulia Vismara.

**Data curation:** Federica Biassoni, Giulia Vismara.

**Formal analysis:** Federica Biassoni.

**Investigation:** Federica Biassoni, Giulia Vismara.

**Methodology:** Federica Biassoni.

**Project administration:** Federica Biassoni.

**Writing – original draft:** Federica Biassoni, Martina Gnerre.

**Writing – review & editing:** Federica Biassoni, Martina Gnerre.

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
