## [Decision Letter · Decision Letter 0]

8 Oct 2025

Dear Dr. Biassoni,

Thank you for submitting your manuscript to PLOS ONE. After careful consideration, we feel that it has merit but does not fully meet PLOS ONE’s publication criteria as it currently stands. Therefore, we invite you to submit a revised version of the manuscript that addresses the points raised during the review process.

The study presents a technically sound and well-conceived exploration of imaginative vocal training as a means to enhance vocal awareness in non-professional singers. The topic aligns well with PLOS ONE’s focus on methodological rigor and scientific validity rather than perceived novelty. While the reviewer finds the design appropriate and the data robust, the manuscript requires substantial revision in organization, interpretation, and clarity before it can be considered for acceptance.

**Required changes for acceptance** include: (1) reorganization of the results section to improve readability and logical flow; (2) verification and correction of statistical reporting (degrees of freedom, effect sizes, and consistency across analyses); (3) acknowledgment of the low internal consistency for the Representation subscale and its implications; and (4) strengthening of the discussion to provide a coherent theoretical explanation for negative or counterintuitive findings.

**Recommended changes** include: (1) improving the English language for clarity and academic tone; (2) ensuring all tables and figures are properly formatted and referenced; and (3) refining minor procedural details for precision.

Overall, the manuscript has merit and potential for publication once the major structural and interpretive issues are resolved. The reviewer’s suggestions are consistent and should be followed closely to achieve the level of clarity and scientific rigor expected in PLOS ONE.

We look forward to receiving your revised manuscript.

Kind regards,

Ramandeep Kaur

Academic Editor

PLOS ONE

Journal Requirements:

2. In the online submission form, you indicated that “The data are available upon request.”

Additional Editor Comments:

The manuscript addresses an engaging and novel area—how imaginative vocal training influences vocal awareness among non-professional singers. The study is thoughtfully designed, and the analysis is largely sound. The reviewer appreciated the paper’s conceptual grounding and the inclusion of both physiological and perceptual dimensions of vocal awareness.

However, several key aspects need significant refinement before the manuscript can be accepted. Please consider the following points carefully:

Major Revisions Required

Clarify and Strengthen Theoretical Interpretation

The negative or reduced post-training scores in Representation and Self-Regulation need a stronger theoretical framing. You may interpret this as a “conscious incompetence” phase—where participants become more aware of their limitations after training, temporarily lowering perceived control.

Ensure this narrative is consistently integrated in the Discussion.

Address Measurement Reliability

The low Cronbach’s alpha (α = .53) for the Representation subscale must be explicitly acknowledged as a limitation and potential reason for inconsistent results.

Reorganize and Simplify the Results Section

Present results by dependent variable (Representation, Interoception, Self-Regulation) rather than by anatomical focus (Vocal Tract, Diaphragm).

Use tables for chi-square and ANOVA results to improve readability and clarity.

Verify and Correct Statistical Reporting

Double-check degrees of freedom, F-statistics, and partial eta-squared (η²ₚ) values. For example, η²ₚ = .033 indicates a small effect, inconsistent with strong significance claims.

Improve English Language and Flow

Some phrasing reflects translation artifacts. A careful line-by-line proofread or language edit is recommended. For instance:

“the singer is her/his own instrument” → “the singer is their own instrument.”

“on the other .” → “on the other.”

Clarify Procedural Details

Under the Diaphragm awareness protocol, Step 4 (“utter Ciao Aurora again”) seems misplaced—confirm whether this refers instead to maintaining the vowel /i/.

Enhance Discussion and Limitations

Add theoretical rationale for unexpected findings.

Acknowledge the psychometric limitation of the Representation subscale.

Reflect on the potential learning curve effect of imaginative training (initial drop before mastery).

Minor Revisions

Include all relevant tables and figures with clear legends and units.

Maintain consistent terminology for constructs and subscales throughout the text.

Conduct a grammar and style review for academic tone and fluency.

With these revisions, the manuscript will present a much stronger, clearer, and theoretically coherent contribution to the field of vocal pedagogy and vocal awareness research.

Reviewer's Responses to Questions

**Comments to the Author**

1. Is the manuscript technically sound, and do the data support the conclusions?

Reviewer #1: Yes

2. Has the statistical analysis been performed appropriately and rigorously?

Reviewer #1: Yes

3. Have the authors made all data underlying the findings in their manuscript fully available?

Reviewer #1: Yes

4. Is the manuscript presented in an intelligible fashion and written in standard English?

Reviewer #1: Yes

Reviewer #1: This is a well-structured and interesting manuscript on a valuable topic. The research question is clear, the methodology is sound, and the discussion thoughtfully interprets the complex results. The primary areas for improvement are in the clarity of writing, data presentation, and the interpretation of some counter-intuitive results. The manuscript sometimes feels like a direct translation, leading to slightly awkward phrasing that can be polished for a native English-speaking academic audience. Here is a detailed review, structured by section, with overall impressions, specific suggestions, and line-level edits.

Major, Structural Recommendations

1. Clarify the "Vocal Awareness" Construct and its Measurement:

o The introduction does an excellent job defining the three components of vocal awareness (Representation, Interoceptive Awareness, Self-Regulation). However, the results for Representation and Self-Regulation are often negative or counter-intuitive (scores decreased post-training).

o Suggestion: In the discussion, you need a stronger, more coherent explanation for why these scores dropped. Your current explanations are good starting points but need to be woven more confidently into the narrative. For example, the drop in self-regulation could be framed as a "conscious incompetence" phase, where training first makes people aware of their lack of control before they can improve it.

o The low Cronbach's alpha (α = .53) for the Representation subscale is a significant limitation. You must acknowledge this more directly in the limitations section, stating that this specific subscale may not have been a reliable measure, which could explain the null/negative findings.

2. Streamline and Reorganize the Results Section:

o The results section is currently very dense and difficult to follow. The separation into "Vocal Tract" and "Diaphragm" for each measure creates repetition.

o Suggestion: Consider reorganizing the results by dependent variable (Representation, Interoception, Self-Regulation) and then within each, discuss the findings for the vocal tract and diaphragm together. This would make it easier for the reader to see the overall pattern for each construct. For example:

3.1. Vocal Apparatus Representation (then discuss results for both vocal tract and diaphragm).

3.2. Interoceptive Awareness (then discuss results for both).

*3.3. Vocal Self-Regulation* (then discuss results for both).

3. Improve Data Presentation:

o The chi-square results are described in the text in a confusing way. It would be much clearer to present these findings in a table.

o Some statistical notations are incorrect (e.g., η²ₚ = .033 is a very small effect size, which seems inconsistent with a significant p-value; double-check these calculations). Degrees of freedom in F-tests also seem off (e.g., F(1, 59) for a within-subjects variable with 60 participants is correct, but later you have F(1, 57) and F(1, 114), which need to be consistent and accurate).

Section-by-Section Feedback & Line Edits

Introduction

• Line 7: "the singer is her/his own instrument" -> "the singer is their own instrument" (using "they/their" is more modern and inclusive).

• Line 35: "on the other ." -> "on the other."

Method

• Materials:

o The description of the My Voice and Me questionnaire is good, but the low alpha (α=.53) for the Representation subscale is a red flag that must be emphasized in the limitations.

o Line ~230: "action. According to Mercado..." - This seems to be the start of the "Imitative Action" section, but the heading is missing. It should be Imitative Action.

• Procedure:

o The procedure is clear. However, for the diaphragm awareness protocol, step 4 says "utter Ciao Aurora again," which seems like a copy-paste error from the vocal tract protocol. Should this be the "maintain the vowel sound ee" task? Please clarify.

Results

• Vocal Apparatus Representation (Vocal Tract):

o You state there was a decrease in scores, which is "inconsistent with H1." This is a key finding and should be highlighted and explained later in the discussion.

o The chi-square results are text-heavy. Use a table.

• General Note: Consistently report effect sizes (η²ₚ) and ensure they are correct. A value of .033 is a small effect, .15 is medium, .25 and above is large.

Discussion

• Strengthen the Explanation for Negative Findings: Don't just state that self-regulation scores dropped. Provide a theoretical rationale. For example: "The observed decrease in self-regulation scores post-training may reflect a transitional phase in skill acquisition. As participants became more aware of their vocal apparatus through training, they may have also become more conscious of their technical shortcomings, leading to a temporary reduction in perceived self-efficacy and control (i.e., 'conscious incompetence')."

• Limitations Section (within Discussion): This is excellent and very self-critical. It greatly strengthens the paper. Be sure to add the point about the low reliability of the Representation subscale here.

Summary of Actionable Revisions

1. Reorganize the Results for better clarity (by DV, not by anatomical focus).

2. Provide and correctly format all Tables and Figures.

3. Thoroughly proofread the manuscript to correct grammatical errors, typos, and awkward phrasing. The examples below are a start, but a line-by-line edit is recommended.

4. Double-check all statistical values (F-statistics, degrees of freedom, p-values, and especially effect sizes η²ₚ).

5. Bolster the Discussion with more confident and theoretical explanations for the negative and null results.

6. Explicitly address the low reliability of the Representation subscale in the limitations.

**Do you want your identity to be public for this peer review?** For information about this choice, including consent withdrawal, please see our Privacy Policy

Reviewer #1: **Yes: ** Mohammed Elrabie Ahmed

---

## [Author Response · Author response to Decision Letter 1]

21 Nov 2025

• A marked-up copy of your manuscript that highlights changes made to the original version. You should upload this as a separate file labelled 'Revised Manuscript with Track Changes'.

Editor Comments:

The manuscript addresses an engaging and novel area—how imaginative vocal training influences vocal awareness among non-professional singers. The study is thoughtfully designed, and the analysis is largely sound. The reviewer appreciated the paper’s conceptual grounding and the inclusion of both physiological and perceptual dimensions of vocal awareness.

However, several key aspects need significant refinement before the manuscript can be accepted. Please consider the following points carefully:

Major Revisions Required

Clarify and Strengthen Theoretical Interpretation

The negative or reduced post-training scores in Representation and Self-Regulation need a stronger theoretical framing. You may interpret this as a “conscious incompetence” phase—where participants become more aware of their limitations after training, temporarily lowering perceived control.

Ensure this narrative is consistently integrated in the Discussion.

Address Measurement Reliability

The low Cronbach’s alpha (α = .53) for the Representation subscale must be explicitly acknowledged as a limitation and potential reason for inconsistent results.

Reorganize and Simplify the Results Section

Present results by dependent variable (Representation, Interoception, Self-Regulation) rather than by anatomical focus (Vocal Tract, Diaphragm).

Use tables for chi-square and ANOVA results to improve readability and clarity.

Verify and Correct Statistical Reporting

Double-check degrees of freedom, F-statistics, and partial eta-squared (η²ₚ) values. For example, η²ₚ = .033 indicates a small effect, inconsistent with strong significance claims.

Improve English Language and Flow

Some phrasing reflects translation artifacts. A careful line-by-line proofread or language edit is recommended. For instance:

“the singer is her/his own instrument” → “the singer is their own instrument.”

“on the other .” → “on the other.”

Clarify Procedural Details

Under the Diaphragm awareness protocol, Step 4 (“utter Ciao Aurora again”) seems misplaced—confirm whether this refers instead to maintaining the vowel /i/.

Enhance Discussion and Limitations

Add theoretical rationale for unexpected findings.

Acknowledge the psychometric limitation of the Representation subscale.

Reflect on the potential learning curve effect of imaginative training (initial drop before mastery).

Minor Revisions

Include all relevant tables and figures with clear legends and units.

Maintain consistent terminology for constructs and subscales throughout the text.

Conduct a grammar and style review for academic tone and fluency.

With these revisions, the manuscript will present a much stronger, clearer, and theoretically coherent contribution to the field of vocal pedagogy and vocal awareness research.

Dear Editor,

We would like to express our sincere gratitude for the time and care you devoted to overseeing the review process of our manuscript. We especially appreciate your effort in summarizing, integrating, and clarifying the reviewer’ comments, which has been extremely helpful in guiding our revisions.

Following your recommendations, and drawing on the constructive feedback offered by the reviewer, we have carefully and thoroughly revised the manuscript. We carefully considered every comment and implemented the required changes aimed at strengthening the rigor, clarity, and overall contribution of our work. We hope that the revisions meet the expectations outlined and enhance the manuscript in a meaningful way.

Reviewer's Responses to Questions

Comments to the Author

1. Is the manuscript technically sound, and do the data support the conclusions?

Reviewer #1: Yes

2. Has the statistical analysis been performed appropriately and rigorously?

Reviewer #1: Yes

3. Have the authors made all data underlying the findings in their manuscript fully available?

Reviewer #1: Yes

4. Is the manuscript presented in an intelligible fashion and written in standard English?

Reviewer #1: Yes

5. Review Comments to the Author

Reviewer #1: This is a well-structured and interesting manuscript on a valuable topic. The research question is clear, the methodology is sound, and the discussion thoughtfully interprets the complex results. The primary areas for improvement are in the clarity of writing, data presentation, and the interpretation of some counter-intuitive results. The manuscript sometimes feels like a direct translation, leading to slightly awkward phrasing that can be polished for a native English-speaking academic audience. Here is a detailed review, structured by section, with overall impressions, specific suggestions, and line-level edits.

We sincerely thank the reviewer for their thoughtful and encouraging review, as well as for the constructive and detailed suggestions provided, which actually helped us to improve the manuscript. We are pleased that the reviewer found the research question clear, the methodology sound, and the discussion valuable.

Major, Structural Recommendations

1. Clarify the "Vocal Awareness" Construct and its Measurement:

o The introduction does an excellent job defining the three components of vocal awareness (Representation, Interoceptive Awareness, Self-Regulation). However, the results for Representation and Self-Regulation are often negative or counter-intuitive (scores decreased post-training).

o Suggestion: In the discussion, you need a stronger, more coherent explanation for why these scores dropped. Your current explanations are good starting points but need to be woven more confidently into the narrative. For example, the drop in self-regulation could be framed as a "conscious incompetence" phase, where training first makes people aware of their lack of control before they can improve it.

o The low Cronbach's alpha (α = .53) for the Representation subscale is a significant limitation. You must acknowledge this more directly in the limitations section, stating that this specific subscale may not have been a reliable measure, which could explain the null/negative findings.

Thank you for this helpful suggestion, based on which we have made the following changes.

• We revised the Discussion to provide a coherent account of the post-training drops in Representation and Self-Regulation as an early “conscious incompetence” phase and response-shift (criterion tightening) effect. We explicitly link this pattern to the concomitant increase in Interoceptive Awareness, arguing that finer perceptual resolution can temporarily lower self-ratings before behavioral mastery consolidates.

• We agree that Cronbach’s α = .53 is low in absolute terms; however, we note that α is systematically deflated in very short scales, especially when items are non-tau-equivalent. Given that our Representation subscale includes only three, partly heterogeneous items, a modest α is expected and does not, per se, invalidate the measure. We have included such consideration so as to address your important comment.

2. Streamline and Reorganize the Results Section:

o The results section is currently very dense and difficult to follow. The separation into "Vocal Tract" and "Diaphragm" for each measure creates repetition.

o Suggestion: Consider reorganizing the results by dependent variable (Representation, Interoception, Self-Regulation) and then within each, discuss the findings for the vocal tract and diaphragm together. This would make it easier for the reader to see the overall pattern for each construct. For example:

3.1. Vocal Apparatus Representation (then discuss results for both vocal tract and diaphragm).

3.2. Interoceptive Awareness (then discuss results for both).

*3.3. Vocal Self-Regulation* (then discuss results for both).

This is an excellent suggestion, and we have implemented it.

3. Improve Data Presentation:

o The chi-square results are described in the text in a confusing way. It would be much clearer to present these findings in a table.

o Some statistical notations are incorrect (e.g., η²ₚ = .033 is a very small effect size, which seems inconsistent with a significant p-value; double-check these calculations). Degrees of freedom in F-tests also seem off (e.g., F(1, 59) for a within-subjects variable with 60 participants is correct, but later you have F(1, 57) and F(1, 114), which need to be consistent and accurate).

This is an another excellent suggestion, so we incorporated a table with the chi-square results in the Results section.

Regarding the appropriate degrees of freedom for the F-tests involving the within-subject factor (Time) and its interaction with the between-subject factor (Training) in our mixed-model ANOVA are indeed (1, 57). This is calculated as (N - k), where N is the total number of participants (60) and k is the number of between-subject groups (3), resulting in 60 - 3 = 57 degrees of freedom for the error term. We have carefully reviewed the entire manuscript and corrected all F-statistics to reflect the accurate degrees of freedom, ensuring the statistical reporting is now fully consistent with our experimental design.

Section-by-Section Feedback & Line Edits

Introduction

• Line 7: "the singer is her/his own instrument" -> "the singer is their own instrument" (using "they/their" is more modern and inclusive).

• Line 35: "on the other ." -> "on the other."

Method

• Materials:

o The description of the My Voice and Me questionnaire is good, but the low alpha (α=.53) for the Representation subscale is a red flag that must be emphasized in the limitations.

o Line ~230: "action. According to Mercado..." - This seems to be the start of the "Imitative Action" section, but the heading is missing. It should be Imitative Action.

• Procedure:

o The procedure is clear. However, for the diaphragm awareness protocol, step 4 says "utter Ciao Aurora again," which seems like a copy-paste error from the vocal tract protocol. Should this be the "maintain the vowel sound ee" task? Please clarify.

We sincerely thank you for your thorough review and for identifying this critical errors in the methodology section. You are absolutely correct. We have corrected the manuscript accordingly and we have integrated a discussion about the low alpha of the questionnaire in the Limitations section.

Results

• Vocal Apparatus Representation (Vocal Tract):

o You state there was a decrease in scores, which is "inconsistent with H1." This is a key finding and should be highlighted and explained later in the discussion.

o The chi-square results are text-heavy. Use a table.

• General Note: Consistently report effect sizes (η²ₚ) and ensure they are correct. A value of .033 is a small effect, .15 is medium, .25 and above is large.

Thank you for this helpful guidance. We have implemented these changes.

Discussion

• Strengthen the Explanation for Negative Findings: Don't just state that self-regulation scores dropped. Provide a theoretical rationale. For example: "The observed decrease in self-regulation scores post-training may reflect a transitional phase in skill acquisition. As participants became more aware of their vocal apparatus through training, they may have also become more conscious of their technical shortcomings, leading to a temporary reduction in perceived self-efficacy and control (i.e., 'conscious incompetence')."

• Limitations Section (within Discussion): This is excellent and very self-critical. It greatly strengthens the paper. Be sure to add the point about the low reliability of the Representation subscale here.

Thank you for these constructive and insightful suggestions. We have revised the manuscript accordingly to strengthen our discussion and address the points you raised. Furthermore, as said above, we have incorporated your point regarding the limitations of the subscale's low reliability into the Limitations section.

Summary of Actionable Revisions

1. Reorganize the Results for better clarity (by DV, not by anatomical focus).

2. Provide and correctly format all Tables and Figures.

3. Thoroughly proofread the manuscript to correct grammatical errors, typos, and awkward phrasing. The examples below are a start, but a line-by-line edit is recommended.

4. Double-check all statistical values (F-statistics, degrees of freedom, p-values, and especially effect sizes η²ₚ).

5. Bolster the Discussion with more confident and theoretical explanations for the negative and null results.

6. Explicitly address the low reliability of the Representation subscale in the limitation

Thank you for this helpful guidance.

---

## [Editor Report · Decision Letter 1]

27 Nov 2025

Efficacy of imaginative vocal training for enhancing vocal awareness

in non-professional singers

PONE-D-25-35461R1

Dear Dr. Biassoni,

We’re pleased to inform you that your manuscript has been judged scientifically suitable for publication and will be formally accepted for publication once it meets all outstanding technical requirements.

Kind regards,

Ramandeep Kaur

Academic Editor

PLOS ONE
---

## [Editor Report · Acceptance letter]

PONE-D-25-35461R1

PLOS ONE

Dear Dr. Biassoni,

I'm pleased to inform you that your manuscript has been deemed suitable for publication in PLOS ONE. Congratulations! Your manuscript is now being handed over to our production team.

Kind regards,

on behalf of

Dr. Ramandeep Kaur

Academic Editor

PLOS ONE